# The relationship between sleep duration and fruit/vegetable intakes in UK adults: a cross-sectional study from the National Diet and Nutrition Survey

Essra A Noorwali,[1,2] Janet E Cade,[1] Victoria J Burley,[1] Laura J Hardie[3]

[1]Nutritional Epidemiology Group, School of Food Science and Nutrition, University of Leeds, Leeds, UK
[2]Department of Clinical Nutrition, Faculty of Applied Medical Sciences, Umm Al-Qura University, Makkah, Saudi Arabia
[3]Division of Epidemiology and Biostatistics, Leeds Institute of Cardiovascular and Metabolic Medicine, School of Medicine, University of Leeds, Leeds, UK

**Correspondence to**
Essra A Noorwali;
fsean@leeds.ac.uk

## ABSTRACT

**Objectives** There is increasing evidence to suggest an association between sleep and diet. The aim of the present study was to examine the association between sleep duration and fruit/vegetable (FV) intakes and their associated biomarkers in UK adults.

**Design** Cross-sectional.

**Setting** Data from The National Diet and Nutrition Survey.

**Participants** 1612 adults aged 19–65 years were included, pregnant/breastfeeding women were excluded from the analyses.

**Outcome measures** Sleep duration was assessed by self-report, and diet was assessed by 4-day food diaries, disaggregation of foods containing FV into their components was conducted to determine total FV intakes. Sleep duration was divided into: short (<7 hours/day), reference (7–8 hours/day) and long (>8 hours/day) sleep periods. Multiple regression adjusting for confounders was used for analyses where sleep duration was the exposure and FV intakes and their associated biomarkers were the outcomes. Restricted cubic spline models were developed to explore potential non-linear associations.

**Results** In adjusted models, long sleepers (LS) consumed on average 28 (95% CI −50 to −6, p=0.01) g/day less of total FV compared to reference sleepers (RS), whereas short sleepers (SS) consumed 24 g/day less (95% CI −42 to −6, p=0.006) and had lower levels of FV biomarkers (total carotenoids, β-carotene and lycopene) compared to RS. Restricted cubic spline models showed that the association between sleep duration and FV intakes was non-linear (p<0.001) with RS having the highest intakes compared to SS and LS. The associations between sleep duration and plasma total carotenoids (p=0.0035), plasma vitamin C (p=0.009) and lycopene (p<0.001) were non-linear with RS having the highest levels.

**Conclusions** These findings show a link between sleep duration and FV consumption. This may have important implications for lifestyle and behavioural change policy.

## INTRODUCTION

The consumption of fruit and vegetables (FV) has shown to improve overall health[1] and reduce the risk of chronic diseases[2–4]

when 400 grams or more are consumed as recommended by The World Health Organization.[5] Hence, identifying lifestyle factors associated with higher intakes of FV is a public health priority.

The relationship between sleep duration and risk of obesity was reported in a recent meta-analysis with short sleep duration associated with a 45% increased risk of obesity due to several behavioural mechanisms including the reduced intake of FV.[6] Thus, it is essential to study FV consumption in relation to sleep duration.

There are limited studies assessing the association between sleep duration and FV consumption in UK adults using validated and detailed dietary data.[7] To our knowledge, this study is the first to use data that disaggregated foods containing FV into their components which helped in assessing total FV intake. Therefore, this study aims to assess the relationship between sleep duration and daily FV consumption and their associated biomarkers in adults aged 19–65 years using data of the National Diet and Nutrition Survey (NDNS) (years 1–4) that represent the UK population.

## METHODS

### Study population

The NDNS is a government-commissioned rolling programme that started in 1992 to assess diet, nutrient intake and nutritional status of the UK population.[8] This study used combined data from years 1–4 of the rolling programme (2008/2009–2011/2012) for adults aged 19–65 years.[9] Between April 2008 and March 2011, random samples of 21 573 addresses from 799 postcode sectors were drawn from the UK Postcode Address File. Households were selected randomly and within the household either one adult (aged 19 years and over) and one child (aged 1.5–18 years), or one child were randomly selected to participate.[8]

### Dietary records

The NDNS assessed dietary intake using a 4-day estimated diary that included instructions on how to complete the diary, as described in detail elsewhere.[8] Participants were asked to record food and drinks consumed both at home and away from home for four consecutive days. Participants were asked to record portion sizes as instructed or in household measures. They were asked to record brand names, ingredients and quantities, cooking methods, leftovers and dietary supplements. Dietary intake was calculated by trained coders and editors in the Diet In Nutrients Out dietary assessment system which calculates food and beverage nutrient intake based on data for >6000 foods. Detailed information on data coding is provided elsewhere.[8]

### Fruit and vegetable intake

To determine the total intakes of FV, disaggregation of foods containing FV into their components was conducted by NDNS. FV content of soft drinks, confectionery, cakes (including fruit cake) and biscuits, sugar preserves (including jam) and sweet spreads, savoury snacks and ice cream were excluded from the estimates because they fell into the high fat/high sugars segment of the eat well plate.[10] The disaggregation process and the calculation of five-a-day portions using disaggregated data is described elsewhere.[8 11]

### Blood sampling (FV biomarkers)

Samples were collected between February 2008 and July 2012; years 1–4 of the NDNS Rolling Programme. In year 1, there was a 2-week time lag between the start of the interviewer and nurse stages. From year 2 onwards, the gap was extended, to an average of 8 weeks, with the aim of increasing nurse stage response rates. Participants were asked a series of screening questions prior to venepuncture to assess their eligibility for giving a blood sample. Participants with a bleeding or clotting disorder or those taking anticoagulant medications were excluded from providing a blood sample. The blood taking procedures including collection, processing, analysing and quality control of the blood samples are explained in further detail elsewhere.[8] This study considered available biomarker measurements related to FV consumption namely plasma vitamin C, total carotenoids, α-carotene, β-carotene and lycopene. The detailed procedure for vitamin C and carotenoid analyses is described elsewhere.[8]

### Sleep duration

Participants were asked about sleep duration in the following form for week nights and weekends by using a computer-assisted personal interview programme:

> Over the last seven days, that is since (date) how long did you usually sleep for on weeknights, that is, Sunday to Thursday nights?

> And over the last seven days, how long did you usually sleep for on a weekend that is Friday and Saturday nights?

An average time per night was sought and if respondents worked on night shifts during the last 2 weeks/weekends, the average time slept during the day should be entered. For this study, two separate variables were generated for sleep duration based on weekdays and weekends for all adults aged 19–65 years from years 1 to 4 in NDNS. Average sleep duration for weekdays and weekends was calculated using the following equation ((minutes slept during the week×5) + (minutes slept during weekends×2))/7. Sleep duration was categorised based on the literature[12–14] to short sleepers (SS) (<7 hours (420 min)), reference sleepers (RS) (7–8 hours ($\geq$420 min and $\leq$480 min)) and long sleepers (LS) (>8 hours (>480 min)).

### Statistical analyses

Descriptive statistics such as means and proportions were conducted to describe adults from the NDNS according to sleep duration categories. P values of <0.05 represent statistical significance. Multiple regression analyses was used to assess the relationship between sleep duration, FV intakes and biomarkers. Model 1 included adjustment for age and sex only whereas model 2 was adjusted for potential confounders that were identified after the development of a directed acyclic graph these were age, sex, socioeconomic status assessed by National Statistics Socioeconomic Classification including eight categories,[15] smoking status[16–19] (current, ex-smoker and never), ethnicity (white, non-white) and energy intake from food. In all analyses, sleep duration was used as the exposure and FV intakes and biomarkers were the outcomes.

We used restricted cubic splines to model non-linear relationships between sleep duration as a continuous exposure (hour/day) and total FV intakes as the outcomes (grams/day). The splines comprised two polynomial segments seperated by three knots (at the following percentiles of sleep duration 10, 50 and 90 as recommended by Harrell[20]) with linear regions before the first knot and after the last.

Sensitivity analyses were conducted including: 1) considering weekdays and weekends separately; and separate analyses were conducted after 2) excluding

participants who consumed vitamins, minerals or/and supplements in the previous year (526 participants); 3) excluding those who self-reported currently having a long-standing illness (see online supplementary material for included illnesses) (547 participants); 4) excluding those taking prescribed medicines (566 participants) 5) excluding those who reported being vegetarian (39 participants) 6) including body mass index (BMI) and physical activity as an additional adjustment to the potential confounders in model 2 and 7) stratifying the analyses between sleep duration and FV intakes by body mass index (BMI). Statistical analyses were conducted using IC Stata V.13,[21] missing data were automatically dropped.

## RESULTS

General characteristics of NDNS adult participants aged 19–65 years according to sleep duration category are shown in table 1. Eighty participants were excluded from the analyses due to lack of sleep data or pregnancy/breast feeding (figure 1). The 1612 adults included in the study had a mean age of 43 years (95% CI 43 to 44) and a mean BMI of 25 kg/m$^2$ (95% CI 25 to 26). Thirty-three per cent (n=539) of the participants were SS, 49% of the participants (n=788) were RS and 18% (n=285) of the participants were LS. In total, 57% (95% CI 55% to 60%) of the participants were female, 90% (95% CI 89% to 92%) were white, 46% (95% CI 43% to 49%) reported taking prescribed medicines and 54% (95% CI 52% to 57%) never smoked.

**Table 1** General characteristics of adults from the NDNS years 1 to 4 according to sleep duration category

| | <7 Hours/day (SS) | Sleep categories 7–8 hours/day (RS) | >8 Hours/day (LS) | Total |
|---|---|---|---|---|
| Observations (n) | 539 | 788 | 285 | 1612 |
| Characteristics | Mean (95% CI) | Mean (95% CI) | Mean (95% CI) | Mean (95% CI) |
| Age (years) | 45 (44 to 46) | 44 (43 to 45) | 39 (38 to 40) | 43 (43 to 44) |
| BMI | 26 (25 to 27) | 25 (25 to 26) | 24 (23 to 25) | 25 (25 to 26) |
| Food energy | 1712 (1665 to 1758) | 1769 (1731 to 1807) | 1645 (1587 to 1703) | 1727 (1701 to 1752) |
| Equivalised household income | 33 000 (31 000 to 35 000) | 34 000 (32 000 to 36 000) | 29 000 (26 000 to 32 000) | 33 000 (32 000 to 34 000) |
| Fruit (g/day) | 98 (89 to 106) | 115 (107 to 124) | 82 (73 to 92) | 103 (98 to 108) |
| Vegetables (g/day) | 178 (168 to 188) | 194 (187 to 201) | 168 (157 to 180) | 185 (180 to 190) |
| Total FV (g/day) | 276 (261 to 291) | 309 (297 to 322) | 250 (233 to 267) | 287 (279 to 296) |
| Plasma vitamin C (µmol/L) | 48 (45 to 51) | 53 (51 to 55) | 56 (53 to 59) | 51 (50 to 53) |
| Plasma total carotenoids (µmol/L) | 2.2 (2.1 to 2.4) | 2.5 (2.4 to 2.7) | 2.4 (2.2 to 2.7) | 2.4 (2.3 to 2.5) |
| Plasma lycopene (µmol/L) | 0.62 (0.57 to 0.67) | 0.73 (0.69 to 0.77) | 0.69 (0.61 to 0.76) | 0.69 (0.66 to 0.72) |
| | % (95% CI) | % (95% CI) | % (95% CI) | % (95% CI) |
| Sex (female) | 55 (51 to 59) | 56 (52 to 59) | 64 (58 to 69) | 57 (55 to 60) |
| Ethnicity (white) | 92 (90 to 94) | 89 (86 to 91) | 88 (84 to 91) | 90 (89 to 92) |
| Has long-standing illness (yes) | 37 (33 to 41) | 30 (26 to 32) | 39 (33 to 45) | 34 (32 to 36) |
| Taking prescribed medicine (yes) | 47 (42 to 51) | 43 (38 to 46) | 53 (46 to 60) | 46 (43 to 49) |
| Employer (full-time or part-time employment) | 68 (64 to 72) | 72 (68 to 75) | 62 (56 to 67) | 69 (67 to 71) |
| SES (lower managerial and professional) | 28 (24 to 32) | 28 (25 to 31) | 22 (17 to 27) | 27 (24 to 29) |
| Smoking (never) | 51 (46 to 55) | 57 (53 to 60) | 54 (48 to 60) | 54 (52 to 57) |
| Consuming five or more portions of FV/day (yes) | 28 (24 to 32) | 35 (31 to 38) | 25 (21 to 31) | 31 (29 to 33) |
| Vegetarian (yes) | 2 (1 to 3) | 3 (2 to 4) | 0.7 (0.1 to 2) | 2 (1 to 3) |
| Has one child aged between 0 and 4 years | 15 (12 to 18) | 13 (10 to 15) | 12 (8 to 16) | 13 (12 to 15) |
| Frequency of drinking alcohol in past 12 months (once or two times in a week or month) | 45 (40 to 49) | 48 (44 to 51) | 50 (44 to 55) | 47 (45 to 50) |

BMI, body mass index; FV, fruit and vegetables; LS, long sleepers; NDNS, National Diet and Nutrition Survey; RS, reference sleepers; SES, socioeconomic status; SS, short sleepers.

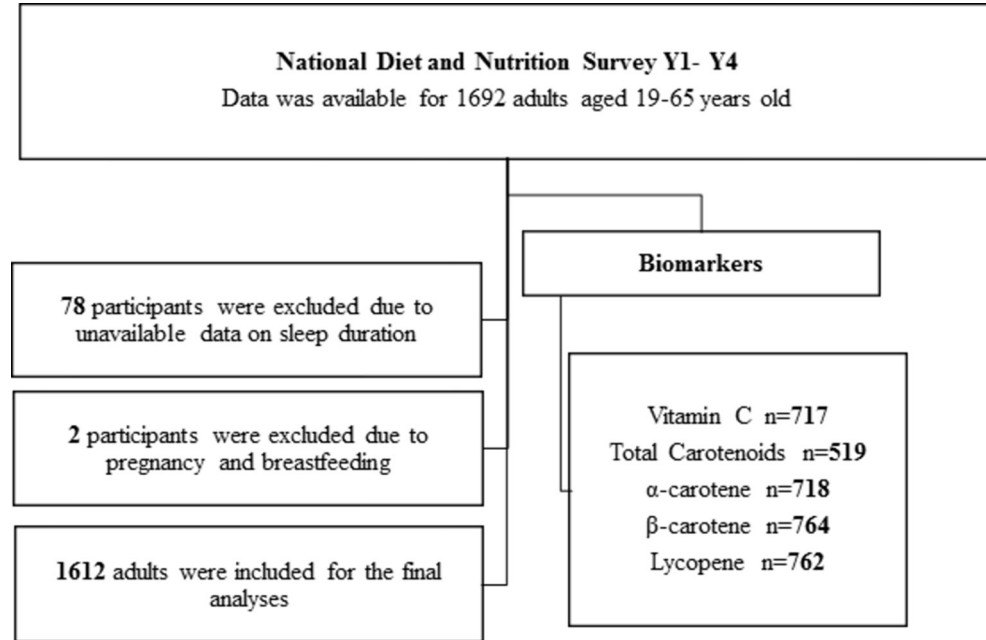

**Figure 1** Participants' flow chart.

Concerning FV consumption, 35% (95% CI 31% to 38%) of RS consumed five or more portions/day of FV whereas 25% (95% CI 21% to 31%) of LS and 28% (95% CI 24% to 32%) of SS consumed five or more portions of FV/day. LS consumed a mean of 250 (95% CI 233 to 267) g/day of total FV, RS consumed a mean of 309 (95% CI 297 to 322) g/day of total FV whereas SS had a mean intake of 276 (95% CI 261 to 291) g/day of total FV (table 1).

In adjusted analyses (model 2), SS and LS ate less fruit (g/day), FV portions and total FV (g/day) compared with RS (table 2). SS ate on average 13 g/day (95% CI −24 to −2, p=0.01) less total fruit, 0.2 (95% CI −0.5 to −0.06, p=0.01) less portions/day of FV and 24 g/day (95% CI −42 to −6, p=0.006) less total FV compared with RS. LS consumed on average 16 g/day (95% CI −30 to −2, p=0.01) less total fruit, 0.2 (95% CI −0.5 to 0.01, p=0.06) less portions/day of FV and 28 g/day (95% CI −50 to −6, p=0.01) less total FV compared with RS. In model 1, SS had on average 17 g/day (95% CI −29 to −5, p=0.004) and LS had on average 19 g/day (95% CI −34 to −4, p=0.009) less vegetable intake compared with RS, but the differences became borderline significant with further adjustment.

In adjusted analyses (model 2), SS had lower levels of plasma FV biomarkers except α-carotene and vitamin C compared with RS. In contrast, LS had higher vitamin C levels compared with RS. LS had 4 µmol/L higher plasma vitamin C (95% CI 0.1 to 8, p=0.04) compared with RS. SS had 0.2 µmol/L lower plasma total carotenoids (95% CI −0.4 to −0.08, p=0.004), 0.05 µmol/L lower plasma β-carotene (95% CI −0.1 to −0.009, p=0.01) and 0.08 µmol/L lower plasma lycopene (95% CI −0.1 to −0.02, p=0.005) compared with RS. This was confirmed with SS having less intake of tomatoes compared with RS in adjusted models (−5 g/day, 95% CI −9 to −0.1, p=0.04). SS had a mean intake of 42 g/day (95% CI 38 to 46) of tomatoes,

RS had 48 g/day (95% CI 45 to 51) and LS had 41 g/day (95% CI 36 to 46).

Restricted cubic spline modelling (figure 2) showed that the association between sleep duration and total FV intake (g/day) was non-linear (p<0.001) with participants sleeping 7–8 hours/day having the highest intakes compared with SS and LS. Similarly, the association between sleep duration and plasma vitamin C (p=0.009) (figure 3A), total carotenoids (p=0.0035) (figure 3B) and lycopene (p<0.001) (figure 3C) were non-linear.

### Sensitivity analyses

Sensitivity analysis showed broadly similar results (available as online supplementary tables 1–7). Including adjustment for BMI and physical activity in the fully adjusted model did not affect the results. Results of separate analyses excluding participants who consumed minerals, vitamins and/or food supplements, being vegan/vegetarian, having a long-standing illness and consuming prescribed medicines, remained similar with SS consuming less FV in comparison to RS but no difference between LS FV intakes and RS. The associations between sleep duration and biomarkers were similar with SS having lower levels compared with RS and LS having higher levels of plasma vitamin C compared with RS. Results dividing the exposure into weekday and weekend sleep duration were similar, SS on average consumed less g/day of FV and had lower levels of biomarkers on weekdays and weekends compared with RS. LS on average consumed less g/day of FV on weekdays compared with RS.

### DISCUSSION

To our knowledge, this is the first nationally representative study to examine the association between

**Table 2** The association between sleep duration categories, FV intakes and their biomarkers of adults from the NDNS years 1–4

| Models | Model 1 (n=1612) | | | | Model 2 (n=1610) | | | |
|---|---|---|---|---|---|---|---|---|
| | SS (<7hours/day) | | LS (>8hours/day) | | SS (<7hours/day) | | LS >8hours/day | |
| | Mean difference (95% CI) | P values | Mean difference (95% CI) | P values | Mean difference (95% CI) | P values | Mean difference (95% CI) | P values |
| **FV intake** | | | | | | | | |
| Total fruit* (g/day) | −19 (−31 to −8) | 0.001 | −24 (−38 to −10) | 0.001 | −13 (−24 to −2) | 0.01 | −16 (−30 to −2) | 0.01 |
| Total vegetable† (g/day) | −17 (−29 to −5) | 0.004 | −19 (−34 to −4) | 0.009 | −10 (−21 to 0.5) | 0.06 | −11 (−25 to 2) | 0.09 |
| FV portions‡ | −0.4 (−0.6 to −0.2) | <0.001 | −0.5 (−0.8 to −0.1) | 0.001 | −0.2 (−0.5 to −0.05) | 0.01 | −0.2 (−0.5 to −0.01) | 0.04 |
| 5-a-day portions§ | −0.4 (−0.7 to −0.2) | <0.001 | −0.5 (−0.8 to−0.1) | 0.002 | −0.2 (−0.5 to−0.06) | 0.01 | −0.2 (−0.5 to 0.01) | 0.06 |
| Total FV¶(g/day) | −37 (−56 to −18) | <0.001 | −44 (−67 to −20) | <0.001 | −24 (−41 to −6) | 0.006 | −28 (−50 to −6) | 0.01 |
| **Nutrients (mg/dL)** | | | | | | | | |
| Vitamin C diet only | −9 (−16 to −2) | 0.01 | −10 (−19 to −1) | 0.02 | −5 (−12 to 1) | 0.1 | −4 (−12 to 4) | 0.3 |
| Vitamin C ** | −13 (−27 to 0.8) | 0.06 | −21 (−39 to −3) | 0.01 | −7 (−21 to 6) | 0.2 | −12 (−30 to 5) | 0.1 |
| **Biomarkers (µmol/L)** | | | | | | | | |
| Vitamin C | −4 (−8 to −1) | 0.006 | 3 (−1 to 7) | 0.1 | −2 (−6 to 0.3) | 0.07 | 4 (0.1 to 8) | 0.04 |
| Total carot†† | −0.2 (−0.4 to −0.1) | 0.002 | −0.1 (−0.3 to 0.1) | 0.3 | −0.2 (−0.4 to −0.08) | 0.004 | −0.06 (−0.2 to 0.1) | 0.5 |
| α-carotene | −0.01 (−0.02 to 0.003) | 0.1 | −0.009 (−0.02 to 0.007) | 0.2 | −0.006 (−0.01 to 0.007) | 0.3 | −0.004 (−0.02 to 0.01) | 0.6 |
| β-carotene | −0.07 (−0.1 to −0.02) | 0.003 | −0.01 (−0.07 to 0.05) | 0.7 | −0.05 (−0.1 to −0.009) | 0.01 | 0.009 (−0.05 to 0.07) | 0.7 |
| Lycopene | −0.1 (−0.1 to −0.04) | 0.001 | −0.06 (−0.1 to 0.01) | 0.09 | −0.08 (−0.1 to −0.02) | 0.005 | −0.05 (−0.1 to 0.02) | 0.1 |

Model 1 adjusted for age and sex.

Model 2 adjusted for age, sex, socioeconomic status, smoking, ethnicity and food energy.

*Total fruit (not including juice)=fruit (g)+dried fruit (g)+smoothie fruit (g).

†Total vegetables=beans (g)+Brassicaceae (g)+other vegetables (g)+tomatoes (g)+tomato puree (g)+yellow red green (g).

‡FV portions=(Fruit (g)+dried fruitx3_mean+tompureex5 mean+beans max mean+Brassicaceae (g)+yellow red green (g)+other vegetables (g)+tomatoes (g))/80.

§5-a-day portions(portions/day)=FV portions+fruit juice portions+smoothie fruit portions.

¶Total FV (not including juice)=total fruit+total vegetables.

**Vitamin C including supplements.

††Total carotenoids=Lutein+ alpha-cryptoxanthin+beta-cryptoxanthin+lycopene+alpha-carotene+beta-carotene.

FV, fruits and vegetables; LS, long sleepers; NDNS, National Diet and Nutrition Survey; SS, short sleepers.

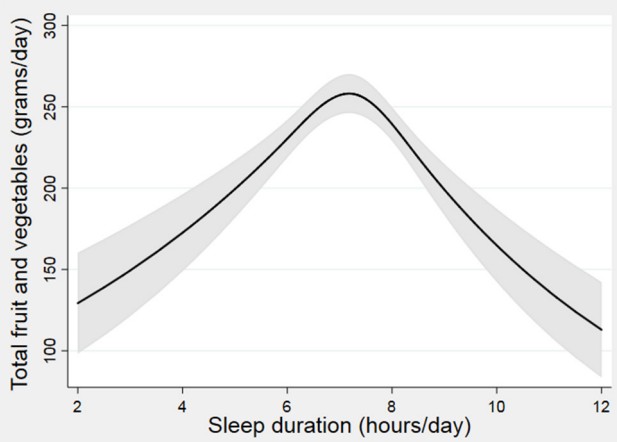

**Figure 2** The association between sleep duration and fruit/vegetable (FV) intakes from the restricted cubic spline modelling. Black lines plot the predicted FV intakes with 95% CI (grey-shaded area) for typical participants (females, white, never smokers, lower managerial and professional occupation, using mean age (43.1) and mean food energy (1727.05)).

sleep duration and FV intakes using disaggregated data among UK adults. The results of this study show SS and LS have lower intakes of FV compared with RS.

Results of FV biomarkers show lower levels of all plasma biomarkers except α-carotene and vitamin C in SS compared with RS in contrast to plasma vitamin C levels in LS which were higher than RS. Similar results were noted after further adjustment for BMI and physical activity, excluding participants who had a long-standing illness, consumed prescribed medicines and those who consumed supplements, minerals or/and vitamins in the previous year. The associations between sleep duration, FV intake and biomarkers were non-linear with RS having the highest intakes and levels of biomarkers compared with SS and LS as shown in the restricted cubic spline modelling. Thus, these findings suggest that among UK adults RS have the highest intakes of FV compared with SS and LS.

These results are in line with several other cross-sectional studies.[12 13 22–24] Although the studies differed in sample size, ethnicity, dietary assessment methods and categorisation of sleep duration, the results showed a lower intake of FV in SS/LS compared with RS. Women with short and long sleep durations had low intakes of FV in the USA or Puerto Rico[22] which was similar to the results of this study. Additionally, short sleep duration was associated with obesity-related behaviours including low FV consumption in rural communities in Missouri,

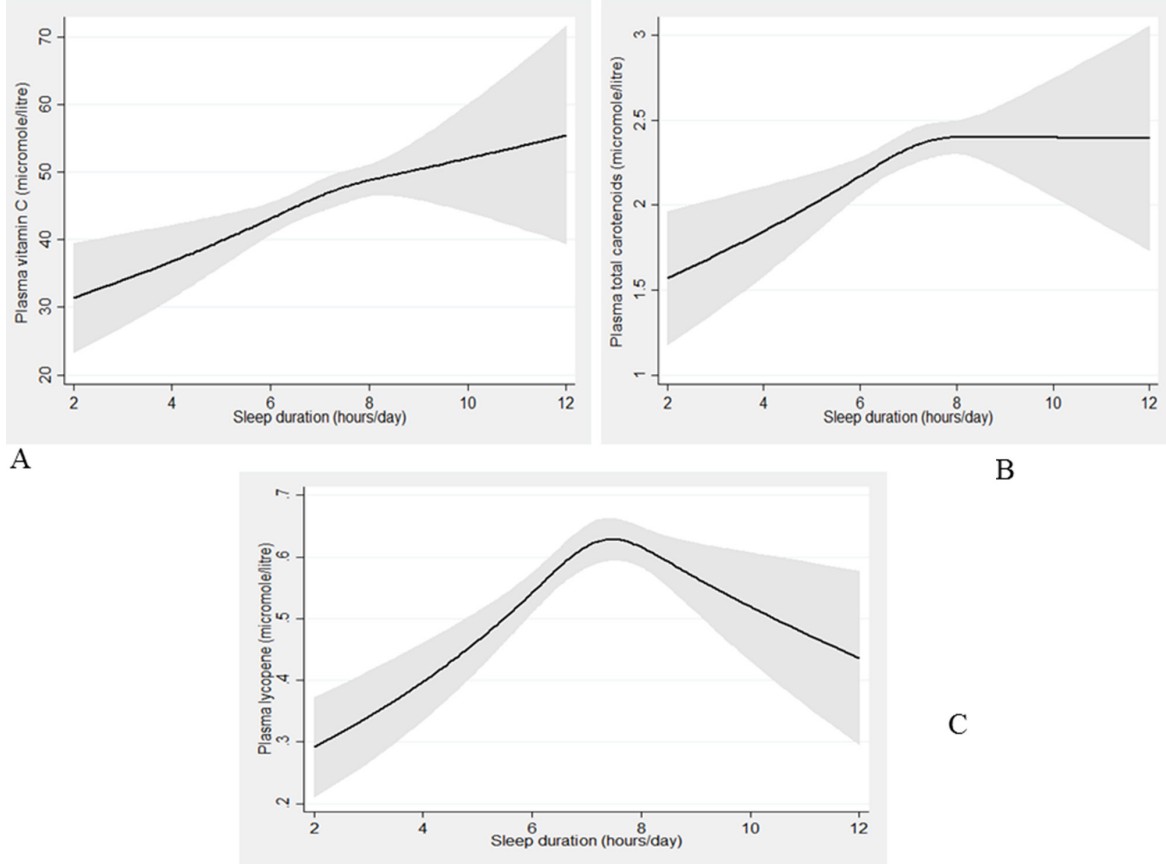

**Figure 3** The association between sleep duration and fruit/vegetable (FV) biomarkers from the restricted cubic spline modelling. Black lines plot the predicted FV biomarkers values: (A) vitamin C, (B) total carotenoids, (C) lycopene with 95% CI (grey-shaded area) for typical participants (females, white, never smokers, lower managerial and professional occupation, using mean age (43.1) and mean food energy (1727.05)).

Tennessee and Arkansas.[23] In a study that examined the association between sleep duration and diet quality among women following childbirth, short sleep duration was not associated with diet quality whereas long sleep duration was associated with lower consumption of total and whole fruit.[25] Katagiri et al[26] measured the association between sleep quality and diet and noted that poor sleep quality was significantly associated with low intakes of total vegetables, green/yellow vegetables and other vegetables.[26] The study suggested that the relationship of dietary intake with sleep quality is similar to that with sleep duration. Regarding the results of FV biomarkers, our results were supported by two other studies.[12 27] Grandner et al[27] showed a significant lower intakes of lycopene in very SS (<5 hours).[27] Beydoun et al[12] reported that short sleep duration was associated with lower serum levels of vitamin C, total carotenoids, α-carotene and β-carotene compared with RS.[12] It is unclear why the results of this study observed a higher plasma vitamin C levels in LS, however, this may be explained by differences in food variety or misreporting of diet intake.[27] This also could be due to biomarkers measuring long-term dietary intake while diet intake was assessed by a 4-day food diary.

In contrast, a recent study examined the association between sleep duration and cardiovascular risk behaviours using data from the UK biobank found results which were inconsistent with our own. Long sleep duration was positively associated with vegetable intake.[28] These results which contrast with our findings may be due to the different assessment methods of sleep duration and FV intake. Sleep duration assessment in the UK biobank did not consider weekday and weekend sleep duration separately as conducted in the NDNS study since sleep duration may differ between those days.[29] Furthermore, self-report of sleep duration may differ by question format by reporting short sleep duration when asked a single question.[30] Sleep duration in the UK biobank study was assessed by asking one question in regard to sleep every 24 hours whereas our study assessed sleep duration by asking two separate questions of sleep based on weeknights and weekends. FV intakes were assessed differently in the UK biobank[28] and the NDNS.[11] Patterson et al[28] assessed FV intake by considering diet intake in the previous year and asking how many pieces of fresh fruit would participants eat per day and how many heaped table-spoons of vegetables participants would eat on average per day. This method was based on the UK guidelines that a portion of vegetables is three heaped tablespoons whereas the NDNS assessed dietary intake using a 4-day estimated diary and disaggregated foods containing FV which is considered a better estimate of average intakes compared with other dietary assessment methods.[31] Additionally, this study conducted supportive biomarker analyses. In a home-based intervention study that assessed the effects of extended bedtimes on sleep duration and food desire, desire for FV was not affected by added sleep.[32] However, the study had several limitations including a small sample size which may limit the generalisability to more diverse populations. One of the major limits of the intervention study[32] is the short duration of intervention (2 weeks) which does not measure the potential effects over a longer period. Experimental studies differ from free-living individuals; therefore, it is required to consider the potential for non-representative samples taking part in experimental studies. Furthermore, the association between FV intake and sleep duration was assessed among American pregnant women. Total daily FV consumption was not associated with sleep duration.[33] This could be due to the different sample and dietary assessment methods. FV was assessed by asking women how many times per day, week or month they consumed FV.

Several potential mechanisms may underlie the association between sleep duration and diet intake.[7 34 35] Short sleep duration or disrupted sleep may lead to emotional stress, impaired decision-making, and increased reward sensitivity to calorie-dense foods and lower FV intake. Changes in appetite hormones, ghrelin and leptin, due to lack/disrupted sleep may increase the preference for energy-dense foods leading to lower intakes of FV. Although potential mechanisms were not measured in this study, they may be the underlying reasons for decreased intake of FV in SS and LS. On the other hand, sleep may be promoted by foods such as kiwifruits, tart cherries, milk and chamomile tea for their impact on tryptophan availability and the synthesis of serotonin and melatonin.[36] This provides insight to the relationship between sleep and diet being potentially bidirectional. Future interventional trials are required to incorporate objective measures of sleep to clarify the relationships between sleep and FV intakes. Sleep extension intervention has been reported to reduce the intakes of free sugars in a 4-week randomised controlled pilot trial.[37] Longer term, fully powered sleep extension studies on FV intake and their associated biomarkers are needed to confirm these results.

### Strengths and limitations of the study

The main strength of this study was the disaggregation of foods containing FV into their components which helped in assessing total FV intake.[11] Furthermore, the 4-day estimated diary has been validated against several biomarkers and demonstrated better estimates of average intakes compared with other dietary assessment methods.

This study has several limitations including the self-report of sleep duration which was based on memory and could cause over-reporting.[38] Further limitations include lack of consideration of other sleep factors such as sleep quality,[26] sleep timing,[39] sleep problems, typical week information, shift-work and chronotype.[28] In year 1, weekend days were oversampled and in year 2, they were undersampled to redress that however, in the years 1–4 combined data there still remains a slightly higher proportion of weekend days. Eating habits vary between weekdays and weekends[40] which could lead to a bias in the reporting of FV intake. The small number

of participants in the obtained biomarkers was a further limitation. The association between sleep duration and FV intake is a bidirectional relationship and the causal pathways underlying the relationship cannot be detected in cross-sectional studies.[26]

## PUBLIC HEALTH IMPLICATIONS

Sleep duration among UK adults has been declining recently with 70% of UK adults sleeping less than 7 hours/night according to the Sleep Council.[41] Additionally, the intake of FV is decreasing among UK adults with only 30% of them meeting the 5-a-day recommendation according to the NDNS results provided by Public Health England.[8] If the results of this study were confirmed in prospective and interventional studies this would highlight the importance of translating the scientific evidence focusing on the relationship between sleep and diet into practical messages that can help the public to prevent chronic diseases. This would include making different populations aware of the relationship between sleep and diet by providing more information on sleep in national dietary guidelines to enhance healthy lifestyle recommendations. In addition, this information can be incorporated in hospitals to educate healthcare professionals, weight-loss programmes and other programmes targeting improvement in overall health. This information is also essential for those caring for at risk groups such as the elderly and those with chronic diseases.[42]

## CONCLUSIONS

The results of this study suggest a link between sleep duration and FV intake. Sleep duration was non-linearly related to self-reported FV intakes and their associated biomarkers with RS having the highest intakes of FV and levels of associated biomarkers compared with SS and LS. These results may have important implications for lifestyle and behavioural change policy.

**Acknowledgements** We thank the study participants and the UK Data Service.

**Contributors** The authors' contributions are as follows: EAN was the principal investigator and contributed to the study design, data analyses, interpretation of the findings and wrote the manuscript. VJB contributed to the study design, data analyses and interpretation of findings. LJH contributed to the study design, data analyses, interpretation of findings and article revision. JEC contributed to interpretation of findings and article revision. All authors read and approved the final version of the manuscript.

**Funding** EAN is in receipt of a scholarship from Umm Al-Qura University, Makkah, Saudi Arabia. JEC was funded by the UK Medical Research Council grant no: MR/L02019X/1.

**Competing interests** None declared.

**Patient consent** Not required.

**Provenance and peer review** Not commissioned; externally peer reviewed.

**Data sharing statement** All National Diet and Nutrition Survey Data are available online at the UK Data Service website (see https://www.gov.uk/government/collections/national-diet-and-nutrition-survey).

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
