## [Reviewer comments · BMJ Open]

ARTICLE DETAILS

TITLE (PROVISIONAL)	The relationship between sleep duration and fruit/vegetable intake in UK adults: a cross-sectional study from The National Diet and Nutrition Survey.
AUTHORS	Noorwali, Essra; Cade, Janet; Burley, Victoria; Hardie, Laura

VERSION 1 – REVIEW

REVIEWER	Marcela Zambrim Campanini Universidade Estadual de Londrina, Brazil.
REVIEW RETURNED	23-Dec-2017

GENERAL COMMENTS	It was with interest that I read the manuscript entitled “The relationship between sleep duration and fruit/vegetable (FV) intake in UK adults: a cross sectional study from the National Diet and Nutrition Survey”. The aim of this paper was to examine the relation between sleep duration and fruit and vegetables consumption, as well as biomarkers related to it. The novelty of the study, as highlighted by the authors, is the use of disaggregated foods containing FV. However, I have some concerns, as follow: Although the authors comment this issue in the end of the discussion section, one concern is about the directionality of the association. Longitudinal studies have demonstrated the influence of dietary patterns, such as the Mediterranean diet, on sleep duration and quality. A possible explanation for such association lies in the amount of antioxidant substances of certain dietary patterns. Once fruits and vegetables are rich in antioxidants, it would be reasonable to consider the variables of the present study from the opposite directionality. In the introduction section, the authors mention a meta-analysis which suggests that the association between sleep duration and obesity might be partially explained by the intake of FV. Did the authors consider to stratify the analyses by the BMI? Another concern is residual confounding, that is, the association between sleep duration and FV consumption could be due to the relation between sleep and healthy habits, such as physical activity and/or sedentary behaviour, which are also associated with a healthier diet pattern. Although the authors adjusted for BMI in sensitivity analyses, if any of those variables (physical activity and sedentary behaviour) is available, it would be of interest to add them to the models. Please, describe which longstanding illnesses were included to the variable of longstanding illness. Please provide information on observational studies that would support the association between sleep duration and specific foods consumption.
--

	The following reference could be of interest to be commented in the discussion section: Association Between Fruit and Vegetable Consumption and Sleep Quantity in Pregnant Women. Matern Child Health J. 2017 May; 21(5):966-973. DOI: 10.1007/s10995-016-2247-y. Minor suggestions:  1) Regarding the sleep variable, my concern is if the researchers asked the participants if the pasting week was a typical week. 2) With exception of tomatoes consumption, did the authors examine relations between specific types of fruits (such as kiwi) and sleep duration to pinpoint specific foods that may be related to sleep duration?
--	---

REVIEWER	Jayne Woodside Queen's University Belfast, UK
REVIEW RETURNED	23-Dec-2017

GENERAL COMMENTS	The paper describes the analysis of the National Diet and Nutrition Survey, examining the association between fruit and vegetable intake and sleep duration. NDNS is a national dietary survey using robust methodologies within a representative sample of the UK population. The paper is well written, the main limitations (examination of cross-sectional associations and self-reported sleep duration) are described clearly, and the authors have conducted a number of sensitivity analyses to further explore their findings. I have a number of minor comments which can be considered by the authors:  1) The authors included shift workers and treated their daytime sleep hours as night sleeping time for non shift workers. Did the authors consider conducting a further sensitivity analysis excluding shift workers, as the eating behaviours of shift workers may differ considerably from those of non-shift workers, while sleeping patterns and sleep quality may also differ? 2) There are a number of spelling errors, e.g. linear p6 line 38; capitalisation of Biobank p12 line 56 and p13 line 6. 3) The format of the data in table 2 is a little difficult to follow - standardisation of number of decimal places, even within a row for difference and 95% CI would make this easier. 4) There is a clear description of the biomarker results on p12 in the first paragraph of discussion, but this would be better described in this detail within the results section, where it does not come through so clearly. 5) I think the public health implications are a little strong, given the cross-sectional nature of the analysis, so would encourage the authors to refine this a little, perhaps adding "if these results were confirmed in prospective and interventional studies..." 6) Similarly the final line of conclusion regarding the need of interventional trials I think belongs earlier in discussion and could be developed - given the bi-directional relationship discussed what would the intervention being tested in such studies? I would concentrate on the first sentence of conclusion, develop it a little regarding what the authors have shown and reinforce that the association was seen with both self-reported FV and FV biomarker assessment. 7) I would add the n value for the biomarker data to the Participants' flow chart in Figure 1.
---

	8) Figure 3 suggests that the observed associations are rather different for vitamin C and total carotenoids than they are for lycopene and for self-reported intake - should the authors consider this and comment on why they think this might be the case?
--	---

VERSION 1 – AUTHOR RESPONSE

Editor Comments to Author:

- Please complete and include a STROBE checklist, ensuring that all points are included and state the page numbers where each item can be found. The checklist can be downloaded from here:

<http://www.strobe-statement.org/?id=available-checklists>

Thank you for this comment, this important additional information has now been added in line with STROBE guidance.

- Please remove the what is already known/what this study adds box. This is not required for submission to BMJ Open.

This box has now been removed in line with the format of BMJ Open.

Reviewer(s)' Comments to Author:

Reviewer: 1

Reviewer Name: Marcela Zambrim Campanini

Institution and Country: Universidade Estadual de Londrina, Brazil.

Please state any competing interests or state 'None declared': None declared.

Please leave your comments for the authors below

It was with interest that I read the manuscript entitled "The relationship between sleep duration and fruit/vegetable (FV) intake in UK adults: a cross sectional study from the National Diet and Nutrition Survey". The aim of this paper was to examine the relation between sleep duration and fruit and vegetables consumption, as well as biomarkers related to it. The novelty of the study, as highlighted by the authors, is the use of disaggregated foods containing FV.

Thank you for your interest in this manuscript.

However, I have some concerns, as follow:

Although the authors comment this issue in the end of the discussion section, one concern is about the directionality of the association. Longitudinal studies have demonstrated the influence of dietary patterns, such as the Mediterranean diet, on sleep duration and quality. A possible explanation for such association lies in the amount of antioxidant substances of certain dietary patterns. Once fruits and vegetables are rich in antioxidants, it would be reasonable to consider the variables of the present study from the opposite directionality.

We appreciate your comment on the directionality of the association however, sleep duration was chosen as the exposure due to several reasons including limited studies considering sleep duration as the exposure, a second reason are the mechanistic studies that support the association between sleep duration and diet intake (mentioned in the discussion section). The rationale and references to the supporting literature are outlined in the introduction and we have also added some further very recent studies to support this. Unfortunately, it was not possible to run a reverse direction analyses due to the unavailable data on specific fruits/vegetables in this dataset. This would therefore be an interesting study to conduct using a different data set.

In the introduction section, the authors mention a meta-analysis which suggests that the association between sleep duration and obesity might be partially explained by the intake of FV. Did the authors consider to stratify the analyses by the BMI?

Thank you for this comment. BMI was considered in this study as further adjustment and in response to this comment we have stratified the analyses by BMI (see table 7 in supplementary material page 7)

Another concern is residual confounding, that is, the association between sleep duration and FV consumption could be due to the relation between sleep and healthy habits, such as physical activity and/or sedentary behaviour, which are also associated with a healthier diet pattern. Although the authors adjusted for BMI in sensitivity analyses, if any of those variables (physical activity and sedentary behaviour) is available, it would be of interest to add them to the models.

We thank the reviewer for this insightful point. In response, we have now added physical activity to the model (see table 5 in supplementary material page 5). We have included a comment on this in the results/discussion sections (page 10-11). Unfortunately, sedentary behaviour was not assessed in the study and so this additional analyses could not be added.

Please, describe which longstanding illnesses were included to the variable of longstanding illness. Thank you for highlighting this. Longstanding illness description has now been added (see supplementary material page 8).

Please provide information on observational studies that would support the association between sleep duration and specific foods consumption.

Thank you for this comment. More observational studies supporting the association between sleep duration and specific food consumption has now been added to the discussion section (page 11-12).

The following reference could be of interest to be commented in the discussion section: Association Between Fruit and Vegetable Consumption and Sleep Quantity in Pregnant Women. *Matern Child Health J.* 2017 May; 21(5):966-973. DOI: 10.1007/s10995-016-2247-y.

This work has been now discussed in the discussion (page 12) and is included in the reference list (reference 33).

Minor suggestions:

1) Regarding the sleep variable, my concern is if the researches asked the participants if the pasting week was a typical week.

Participants were not specifically asked whether the past week was a typical week however, the interviewers were instructed that if participants could not answer due to wide variability of sleep pattern from night to night in that week, to answer "Don't know". However, we recognise this as a potential limitation and have included this in the strengths/limitation section (page 13).

2) With exception of tomatoes consumption, did the authors examine relations between specific types of fruits (such as kiwi) and sleep duration to pinpoint specific foods that may be related to sleep duration?

Thank you for this useful comment. The authors considered examining specific foods in relation to sleep duration however, this raw data was not available for us to analyse which limited us in looking at

specific foods. Individual variables for specific food intakes was unavailable except for tomato that was specifically examined by us to confirm the associated biomarker results (lycopene).

Reviewer: 2

Reviewer Name: Jayne Woodside

Institution and Country: Queen's University Belfast, UK

Please state any competing interests or state 'None declared': None declared

Please leave your comments for the authors below

The paper describes the analysis of the National Diet and Nutrition Survey, examining the association between fruit and vegetable intake and sleep duration. NDNS is a national dietary survey using robust methodologies within a representative sample of the UK population. The paper is well written, the main limitations (examination of cross-sectional associations and self-reported sleep duration) are described clearly, and the authors have conducted a number of sensitivity analyses to further explore their findings. I have a number of minor comments which can be considered by the authors:

1) The authors included shift workers and treated their daytime sleep hours as night sleeping time for non-shift workers. Did the authors consider conducting a further sensitivity analysis excluding shift workers, as the eating behaviours of shift workers may differ considerably from those of non-shift workers, while sleeping patterns and sleep quality may also differ?

Thank you for this insightful point unfortunately, specific questions were not asked regarding shift work in the NDNS. However, interviewers were instructed that if respondents worked on night shifts during the last two weeks to enter the average time slept during the day. In response, we have added this to the limitation section (page 13).

2) There are a number of spelling errors, e.g. linear p6 line 38; capitalisation of Biobank p12 line 56 and p13 line 6.

Thank you for this comment. These changes have now been made.

3) The format of the data in table 2 is a little difficult to follow - standardisation of number of decimal places, even within a row for difference and 95% CI would make this easier.

Thank you for this comment. This has now been re-formatted accordingly in table 2 and the sensitivity analyses tables.

4) There is a clear description of the biomarker results on p12 in the first paragraph of discussion, but this would be better described in this detail within the results section, where it does not come through so clearly.

We agree that this aspect of the results section was not sufficiently clear, it has now been edited accordingly (see page 8)

5) I think the public health implications are a little strong, given the cross-sectional nature of the analysis, so would encourage the authors to refine this a little, perhaps adding "if these results were confirmed in prospective and interventional studies..."

We agree that the previous wording was too strong and have included a sentence as suggested (see page 14)

6) Similarly the final line of conclusion regarding the need of interventional trials I think belongs earlier in discussion and could be developed - given the bi-directional relationship discussed what would the intervention being tested in such studies? I would concentrate on the first sentence of conclusion, develop it a little regarding what the authors have shown and reinforce that the association was seen with both self-reported FV and FV biomarker assessment.

In line with this helpful comment, we have now developed the conclusion and included the need for interventional trials in the discussion (see page 13).

7) I would add the n value for the biomarker data to the Participants' flow chart in Figure 1. This has now been added to the Participants' flow chart and removed from table 2.

8) Figure 3 suggests that the observed associations are rather different for vitamin C and total carotenoids than they are for lycopene and for self-reported intake - should the authors consider this and comment on why they think this might be the case?

The association between sleep duration and plasma vitamin C ($p=0.009$) (Fig 3A), total carotenoids ($p=0.0035$) (Fig 3B) and lycopene ($p<0.001$) (Fig 3C) were non-linear. In response, the following sentence has been added to the discussion section "It is unclear why the results of this study observed a higher plasma vitamin C levels in LS however, this may be explained by differences in food variety or misreporting of diet intake[27]. This also could be due to biomarkers measuring long term dietary intake while diet intake was assessed by a 4-day food diary."

VERSION 2 – REVIEW

REVIEWER	Jayne Woodside Queen's University Belfast, UK
REVIEW RETURNED	25-Jan-2018

GENERAL COMMENTS	The authors have responded to all my queries - many thanks.
---

REVIEWER	Marcela Zambrim Campanini Universidade Estadual de Londrina, Brazil.
REVIEW RETURNED	03-Feb-2018

GENERAL COMMENTS	After reading the authors' responses for each concern raised in the first review, I accept this study for publication. The authors have added crucial points to the discussion section, and have performed interesting analysis in order to respond important questions related to data interpretation. I am convinced that this work brings new information on the association between sleep and food consumption from an epidemiologic perspective.
--